# Are metabolic abnormalities the missing link between complete blood count-derived inflammatory markers and diabetic foot? Evidence from a large population study

Yang Zhang[1]*◔, Shumin Zhou[2]◔, Xianbin Wang[1], Haiyan Zhou[1]

**1** Department of Plastic Surgery, Foresea Life Insurance Guangzhou General Hospital, Guangzhou, Guangdong Province, China, **2** Department of Endocrinology, Guangzhou Development District Hospital, Guangzhou, Guangdong Province, China

◔ These authors contributed equally to this work.

* joshuayangzhang@qq.com

## Abstract

### Background

Diabetic foot is a serious complication of diabetes, and inflammation plays a key role in its pathogenesis. This population-based study investigates associations between complete blood count (CBC)-derived inflammatory markers and diabetic foot, while evaluating metabolic mediators in these relationships.

### Methods

Data from 1,246 participants across three National Health and Nutrition Examination Survey (NHANES) cycles (1999–2004) were analyzed. Calculated inflammatory markers included monocyte-to-lymphocyte ratio (MLR), neutrophil-to-lymphocyte ratio (NLR), neutrophil-monocyte-to-lymphocyte ratio (NMLR), and systemic inflammatory response index (SIRI). Weighted logistic regression models assessed marker-diabetic foot associations, supplemented by subgroup and restricted cubic spline (RCS) analyses for nonlinearity. Mediation analysis quantified metabolic contributions.

### Results

CBC-derived inflammatory markers demonstrated significant positive correlations with diabetic foot risk. Risk increased as quartiles for these markers increased. RCS analysis further revealed a significant nonlinear relationship between them. Serum creatinine (12.46%) and albumin (11.33%) mediated significant proportions of these associations.

**Data availability statement:** All relevant data are within the paper and its Supporting Information files.

**Funding:** This work was supported by Health Science and Technology Project of Guangzhou City, 20241A010119. The funders had no role in study design, data collection and analysis, decision to publish, or preparation of the manuscript.

**Competing interests:** The authors have declared that no competing interests exist.

## Conclusions

CBC-derived inflammatory markers serve as accessible predictors of diabetic foot risk, with nonlinear patterns. The partial mediation by metabolic indicators highlights dual inflammatory-metabolic pathways in diabetic foot pathogenesis. Routine CBC-derived inflammatory markers monitoring could enable early risk stratification, while targeting metabolic abnormalities may amplify preventive strategies.

## Introduction

Diabetic foot stands as one of the most devastating complications of diabetes, imposing immense clinical and socioeconomic burdens through its relentless progression to ulceration, infection, and limb loss. Diabetic foot is characterized by infection, ulceration, or deep tissue destruction associated with neurological abnormalities and peripheral arterial disease. This condition not only diminishes quality of life but also drives catastrophic healthcare costs, with amputation rates and five-year post-amputation mortality exceeding 15% and 50% respectively [1]. Despite advances in medical care, the management of diabetic foot remains challenging due to its multifactorial etiology and high risk of recurrence [2]. Alarmingly, over one-third of diabetic patients develop foot ulcers during their lifetime [3], underscoring the urgent need for early risk stratification tools.

Current approaches to diagnosis and treatment of diabetic foot are limited by multiple factors, including delayed diagnosis, inadequate patient education, and inadequate management of comorbidities. Traditional diagnostic methods mainly focus on clinical examination and imaging studies, which may not fully capture the underlying inflammatory process [4]. Complete blood count (CBC)-derived inflammatory markers, such as neutrophil-to-lymphocyte ratio (NLR) and monocyte-to-lymphocyte ratio (MLR), have become potential indicators of systemic inflammation and are associated with various cardiovascular and renal complications in diabetic patients [5]. Studies have shown that elevated NLR is associated with an increased risk of Diabetic Foot Ulcers(DFUs) and poor wound healing outcomes [6], and higher MLR is associated with a stronger inflammatory response and worse clinical prognosis in patients with diabetes [7]. In addition, the relationship between CBC-derived inflammatory markers and diabetic foot is affected by multiple factors, including demographic characteristics, comorbidities, and lifestyle behaviors. For example, studies have shown that age, gender, and the presence of comorbidities (such as hypertension and cardiovascular disease) influence the inflammatory response, as well as the progression of DFUs [8]. Furthermore, lifestyle factors, such as smoking, physical activity, and dietary habits, have been suggested to play a role in modulating systemic inflammation and risk of diabetes complications [9].

While prior work has identified individual CBC markers as prognostic tools, critical gaps persist. First, existing studies predominantly examine isolated markers (e.g., NLR or MLR) rather than evaluating their combined predictive utility. Second, no population-level research has systematically explored how metabolic

dysregulation—a hallmark of diabetes progression—mediates the relationship between CBC-derived inflammation and foot complications. This oversight is striking given that metabolic abnormalities like hypoalbuminemia and elevated creatinine directly influence both systemic inflammation and microvascular damage. Our study addresses these gaps through three key innovations: (1) simultaneous analysis of four CBC-derived inflammatory indices (MLR, NLR, NMLR, SIRI) in a nationally representative cohort; (2) rigorous quantification of metabolic mediation effects using causal inference methods; and (3) identification of nonlinear risk to guide clinical decision-making.

## Materials and methods

### Study design and population

We analyzed data from three cycles (1999–2004) of the National Health and Nutrition Examination Survey (NHANES), a nationally representative cross-sectional study assessing health and nutrition in non-institutionalized U.S. civilians. NHANES is reviewed and approved by the National Center for Health Statistics (NCHS) Research Ethics Review Board and obtains informed consent from all participants. After excluding participants with invalid diabetic foot assessments (N = 19,652), non-diabetic individuals (N = 8,509), and incomplete baseline data (N = 201), the final cohort included 1,246 adults (117 diabetic foot cases, 1,129 controls) (Fig 1).

### CBC-derived inflammatory markers

Five indices were calculated:

(1) Monocyte - Lymphocyte Ratio (MLR) = Monocyte count/Lymphocyte count

(2) Neutrophil - Lymphocyte Ratio (NLR) = Neutrophil count/Lymphocyte count

(3) Neutrophil and Monocyte - Lymphocyte Ratio (NMLR) = (Monocyte count + Neutrophil count)/Lymphocyte count

(4) Systemic Inflammatory Response Index (SIRI) = (Neutrophil count×Monocyte count)/Lymphocyte count

(5) Derived Neutrophil to Lymphocyte Ratio (dNLR) = Neutrophil count/(White blood cell count – Lymphocyte count)

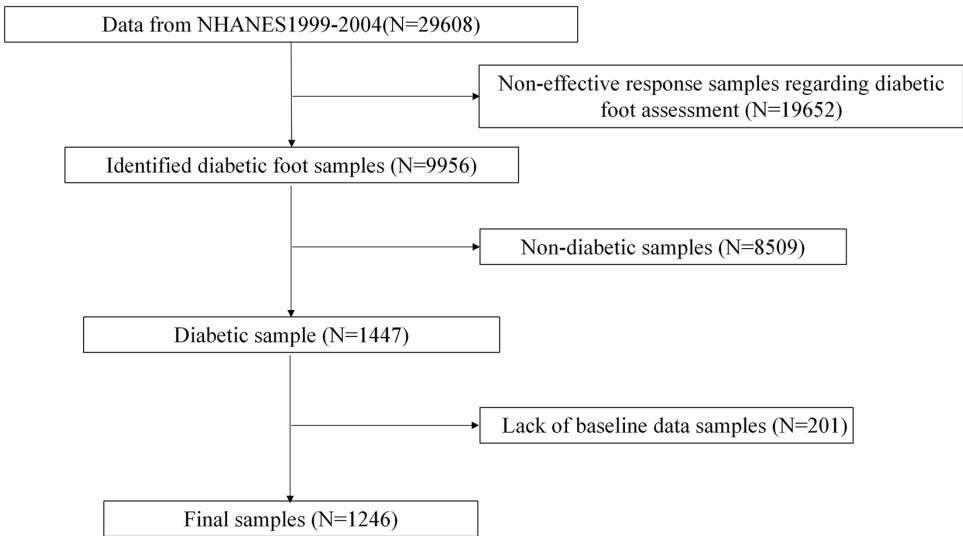

**Fig 1. Participants flowchart.**

### Covariant

Potential confounders were categorized as follows:

(1) Demographics: age, gender, race (Mexican Americans, non-Hispanic Blacks, non-Hispanic Whites, other Hispanics and other), education (five levels: less than 9th grade to college graduate), marital status (married/unmarried), and poverty-income ratio(PIR).

(2) Behavioral: alcohol consumption (drink) (≥12 drinks/year).

(3) Comorbidities: Physician-diagnosed hypertension, coronary heart disease (CHD), heart failure(HF).

(4) Metabolic biomarkers: Total cholesterol(TC), high-density lipoprotein cholesterol(HDL-C), triglyceride(TG), hemoglobin A1c(HbA1c), albumin(ALB), serum creatinine(Cr), uric acid(UA).

(5) Hematologic biomarkers: White blood cells (WBC), neutrophils, lymphocytes, hemoglobin(HB), platelet count(PLT), red blood cell distribution width(RDW).

Sociodemographic and behavioral variables were derived from NHANES demographic modules; comorbidities from self-reported medical histories; metabolic and hematologic biomarker data from standardized laboratory measurements.

### Statistical analysis

Weighted logistic regression evaluated associations between inflammatory markers (continuous and quartile-categorized) and diabetic foot, using three models:

Model 1: Unadjusted.

Model 2: Adjusted for demographics.

Model 3: Model 2+comorbidities (heart failure) and biomarkers (albumin, serum creatinine, RDW).

Subgroup analyses stratified by age, gender, race, marital status, and comorbidities. Restricted cubic splines assessed nonlinear relationships. Mediating effect analysis quantified proportions mediated by creatinine, albumin, and RDW. The existence of the mediating effect is defined by meeting all the following conditions: having a significant indirect effect, a significant total effect, and a proportional mediating effect.

All data processing and analysis were completed using R software (version 4.1.1). The median and interquartile range (IQR) were used to describe continuous variables with a non-normal distribution to analyze baseline characteristics. Categorical variables were reported as sample counts and weighted percentages. To examine the variation in variable characteristics between the CBC-derived inflammatory marker groups (quartiles), we employed the Wilcoxon rank-sum test for continuous variables and the Rao-Scott chi-square test for the weighted percentages of categorical variables, providing a comprehensive description of the entire population. Mediation effect analysis was conducted using "mediation." Statistical analysis was two-sided, with $p < 0.05$ considered statistically significant.

### Results

#### Baseline characteristics

There were differences in the following baseline characteristics between the diabetic foot group and the control group: heart failure(HF) (P=0.037), albumin (ALB) (P<0.001), serum creatinine (Cr) (P<0.001), red blood cell distribution width (RDW) (P=0.004), NLR (P=0.003), MLR (P=0.013), NMLR (P=0.002), and SIRI (P=0.006). At the same time, there were no differences in other baseline indicators between the diabetic foot group and the control group, indicating that these indicators were relatively balanced in both groups, allowing for the exclusion of confounding factors from these baseline indicators in subsequent comparisons. (Table 1).

**Table 1. Baseline characteristics of the study participants.**

| Variable | N | Overall N = 1,246[a] | Diabetic foot N = 117[a] | Control N = 1,129[a] | p-value[b] |
|---|---|---|---|---|---|
| **Gender** | 1,246 | | | | 0.300 |
| Male | | 635 (50.96%) | 65 (55.56%) | 570 (50.49%) | |
| Female | | 611 (49.04%) | 52 (44.44%) | 559 (49.51%) | |
| **Age** | 1,246 | 64.87 (11.55) | 64.74 (11.66) | 64.88 (11.54) | 0.790 |
| **Race** | 1,246 | | | | 0.570 |
| Mexican Americans | | 371 (29.78%) | 37 (31.62%) | 334 (29.58%) | |
| Other Hispanics | | 55 (4.41%) | 6 (5.13%) | 49 (4.34%) | |
| Non-Hispanic Whites | | 488 (39.17%) | 50 (42.74%) | 438 (38.80%) | |
| Non-Hispanic Blacks | | 286 (22.95%) | 22 (18.80%) | 264 (23.38%) | |
| Others | | 46 (3.69%) | 2 (1.71%) | 44 (3.90%) | |
| **Education** | 1,246 | | | | 0.690 |
| Less Than 9th Grade | | 376 (30.18%) | 29 (24.79%) | 347 (30.74%) | |
| 9-11th Grade | | 241 (19.34%) | 26 (22.22%) | 215 (19.04%) | |
| High School | | 249 (19.98%) | 23 (19.66%) | 226 (20.02%) | |
| AA Degree | | 254 (20.39%) | 27 (23.08%) | 227 (20.11%) | |
| College Graduate | | 126 (10.11%) | 12 (10.26%) | 114 (10.10%) | |
| **PIR** | 1,246 | 2.20 (1.45) | 2.04 (1.37) | 2.22 (1.46) | 0.230 |
| **Marry** | 1,246 | 735 (58.99%) | 65 (55.56%) | 670 (59.34%) | 0.430 |
| **Drink** | 1,246 | 695 (55.78%) | 72 (61.54%) | 623 (55.18%) | 0.190 |
| **Hypertension** | 1,246 | | | | 0.640 |
| Hypertension | | 828 (66.45%) | 80 (68.38%) | 748 (66.25%) | |
| Health | | 418 (33.55%) | 37 (31.62%) | 381 (33.75%) | |
| **CHD** | 1,246 | | | | 0.067 |
| CHD | | 148 (11.88%) | 20 (17.09%) | 128 (11.34%) | |
| Health | | 1,098 (88.12%) | 97 (82.91%) | 1,001 (88.66%) | |
| **HF** | 1,246 | | | | 0.037* |
| Heart failure | | 158 (12.68%) | 22 (18.80%) | 136 (12.05%) | |
| Health | | 1,088 (87.32%) | 95 (81.20%) | 993 (87.95%) | |
| **TC (mmol/L)** | 1,246 | 200.66 (43.41) | 196.59 (46.35) | 201.08 (43.10) | 0.220 |
| **HDL-C (mmol/L)** | 1,246 | 48.05 (14.03) | 47.09 (16.07) | 48.15 (13.80) | 0.065 |
| **ALB(g/L)** | 1,246 | 4.15 (0.37) | 4.03 (0.36) | 4.16 (0.37) | <0.001* |
| **TG (mmol/L)** | 1,246 | 198.43 (42.60) | 195.28 (45.70) | 198.75 (42.27) | 0.320 |
| **Cr (umol/L)** | 1,246 | 1.05 (0.90) | 1.38 (1.56) | 1.02 (0.79) | <0.001* |
| **UA(umol/L)** | 1,246 | 5.60 (1.57) | 5.83 (1.66) | 5.58 (1.56) | 0.150 |
| **WBC(%)** | 1,246 | 7.48 (2.21) | 7.81 (2.66) | 7.45 (2.15) | 0.230 |
| **Neutrophils (%)** | 1,246 | 4.50 (1.69) | 4.88 (2.03) | 4.46 (1.64) | 0.069 |
| **Lymphocytes (%)** | 1,246 | 2.14 (0.89) | 2.04 (0.94) | 2.15 (0.89) | 0.075 |
| **HB (g/dL)** | 1,246 | 13.93 (1.59) | 13.62 (1.62) | 13.96 (1.59) | 0.052 |
| **PLT(%)** | 1,246 | 254.50 (74.98) | 262.07 (80.58) | 253.71 (74.37) | 0.290 |
| **RDW(%)** | 1,246 | 13.16 (1.34) | 13.42 (1.27) | 13.13 (1.34) | 0.004* |
| **HbA1c(%)** | 1,246 | 7.48 (1.83) | 7.67 (2.17) | 7.46 (1.79) | 0.640 |
| **NLR** | 1,246 | 2.41 (1.45) | 2.83 (1.74) | 2.37 (1.41) | 0.003* |
| **dNLR** | 1,246 | 0.83 (0.05) | 0.84 (0.05) | 0.83 (0.05) | 0.750 |
| **MLR** | 1,246 | 0.31 (0.15) | 0.35 (0.21) | 0.30 (0.14) | 0.013* |

*(Continued)*

**Table 1.** (Continued)

| Variable | N | Overall | Diabetic foot | Control | p-value[b] |
|---|---|---|---|---|---|
| | | N = 1,246[a] | N = 117[a] | N = 1,129[a] | |
| **NMLR** | 1,246 | 2.72 (1.55) | 3.18 (1.91) | 2.67 (1.50) | 0.002* |
| **SIRI** | 1,246 | 1.41 (1.02) | 1.79 (1.56) | 1.37 (0.94) | 0.006* |

[a] Mean (SD) or Frequency (%)

[b] Pearson's Chi-squared test; Wilcoxon rank sum test; Fisher's exact test

PIR poverty-to-income ratio, CHD coronary heart disease, TC total cholesterol, TG triglyceride, HDL-C high-density lipoprotein cholesterol, ALB albumin, Cr serum creatinine, UA Uric acid, WBC white blood cell count, HB hemoglobin, PLT platelet count, RDW Red blood cell distribution width, HBA1c hemoglobin A1c

## The relationship between CBC-derived inflammatory markers and diabetic foot

Table 2 presented the analysis of the association between CBC-derived inflammatory markers and diabetic foot.

Model 1 presented a positive correlation between MLR, NLR, NMLR, SIRI and diabetic foot when covariates were not included: MLR (OR: 6.17, 95%CI: 2.14–16.95, P<0.001); NLR (OR: 1.17, 95%CI: 1.05–1.29, P=0.003); NMLR (OR: 1.16, 95%CI: 1.06–1.28, P=0.002); SIRI (OR: 1.32, 95%CI: 1.14–1.51, P<0.001). From the data trend, as the indicator values increased from Q1 to Q4, the risk of diabetic foot showed an upward trend, with the "p for trend" values for each indicator being MLR: 0.016, NLR: 0.003, NMLR: 0.005, SIRI: 0.018.

In Model 2, race, gender, age, education, PIR, and marital status were included as covariates. The CBC-derived inflammatory markers remained positively correlated with diabetic foot: MLR (OR: 6.18, 95% CI: 2.04–17.81, P<0.001); NLR (OR: 1.16, 95% CI: 1.04–1.29, P=0.008); NMLR (OR: 1.15, 95% CI: 1.04–1.27, P=0.006); SIRI (OR: 1.29, 95% CI: 1.11–1.49, P<0.001). Similarly, as the indicator values increased from Q1 to Q4, the risk of diabetic foot continued to rise, but the significance of the trend for some indicators had changed. The "p for trend" values for each indicator were as follows: MLR: 0.024, NLR: 0.007, NMLR: 0.012, SIRI: 0.044.

In Model 3, heart failure, ALB, Cr, and RDW were further included as covariates, and it was similarly found that CBC-derived inflammatory markers were positively correlated with diabetic foot: MLR (OR: 4.55, 95%CI: 1.37–14.02, P=0.010); NLR (OR: 1.12, 95%CI: 1.00–1.25, P=0.035); NMLR (OR: 1.12, 95%CI: 1.01–1.24, P=0.030); SIRI (OR: 1.25, 95%CI: 1.06–1.46, P=0.006). From the data trend, the "p for trend" value of NLR was 0.036, indicating that as the indicator values increased from Q1 to Q4, the risk of diabetic foot increased for some indicators, although the overall significance of the trend changed, and the trend for some indicators was no longer significant.

## RCS of CBC-derived inflammatory markers and diabetic foot

The relationship between CBC-derived inflammatory markers and diabetic foot was studied using RCS, adjusting for all relevant covariates. Fig 2 presented that there were statistically significant nonlinear relationships between MLR, NLR, NMLR, and SIRI with diabetic foot.

## Subgroup analysis

S1 Fig. presented the subgroup analysis of CBC-derived inflammatory markers and diabetic foot, conducted through fully adjusted multivariable logistic regression, based on baseline characteristics such as gender, race, marital status, hypertension, coronary heart disease, heart failure, and age. The results indicated that most analyses did not show differences within the groups, with the only exception being a stronger positive correlation between MLR and diabetic foot among participants with coronary heart disease.

**Table 2. The relationship between CBC-Derived inflammatory markers and diabetic foot.**

| Characterisitic | | Model1 | | | Model2 | | | Model3 | | |
|---|---|---|---|---|---|---|---|---|---|---|
| | | OR | 95%CI | P-value | OR | 95%CI | P-value | OR | 95%CI | P-value |
| MLR | MLR continuous | 6.17 | 2.14,16.95 | <0.001* | 6.18 | 2.04,17.81 | <0.001* | 4.55 | 1.37,14.02 | 0.010* |
| | MLR quantile | | | | | | | | | |
| | Q1 (MLR<0.21) | Ref | Ref | | Ref | Ref | | Ref | Ref | |
| | Q2 (0.21≤MLR<0.27) | 0.91 | 0.49,1.66 | 0.750 | 0.90 | 0.49,1.65 | 0.728 | 0.87 | 0.47,1.61 | 0.657 |
| | Q3 (0.27≤MLR<0.36) | 1.17 | 0.66,2.07 | 0.593 | 1.15 | 0.64,2.09 | 0.632 | 1.07 | 0.59,1.95 | 0.828 |
| | Q4 (MLR≥0.36) | 1.78 | 1.06,3.05 | 0.033* | 1.79 | 1.02,3.19 | 0.046* | 1.42 | 0.79,2.59 | 0.243 |
| | p for trend | | | 0.016* | | | 0.024* | | | 0.168 |
| NLR | NLR continuous | 1.17 | 1.05,1.29 | 0.003* | 1.16 | 1.04,1.29 | 0.008* | 1.12 | 1.00,1.25 | 0.035* |
| | NLR quantile | | | | | | | | | |
| | Q1 (NLR<1.56) | Ref | Ref | | Ref | Ref | | Ref | Ref | |
| | Q2 (1.56≤NLR<2.1) | 1.16 | 0.63,2.14 | 0.635 | 1.10 | 0.59,2.06 | 0.763 | 1.10 | 0.59,2.07 | 0.776 |
| | Q3 (2.1≤NLR<2.86) | 1.46 | 0.82,2.65 | 0.199 | 1.39 | 0.77,2.55 | 0.278 | 1.36 | 0.75,2.51 | 0.320 |
| | Q4 (NLR≥2.86) | 2.15 | 1.25,3.78 | 0.006* | 2.04 | 1.16,3.67 | 0.015* | 1.77 | 0.99,3.23 | 0.059 |
| | p for trend | | | 0.003* | | | 0.007* | | | 0.036* |
| NMLR | NMLR continuous | 1.16 | 1.06,1.28 | 0.002* | 1.15 | (1.04,1.27 | 0.006* | 1.12 | 1.01,1.24 | 0.030* |
| | NMLR quantile | | | | | | | | | |
| | Q1 (NMLR< 1.79) | Ref | Ref | | Ref | Ref | | Ref | Ref | |
| | Q2 (1.79≤NMLR< 2.41) | 0.86 | 0.46,1.61 | 0.643 | 0.81 | 0.43,1.53 | 0.514 | 0.76 | 0.40,1.45 | 0.413 |
| | Q3 (2.41≤NMLR< 3.21) | 1.54 | 0.89,2.71 | 0.126 | 1.45 | 0.82,2.61 | 0.200 | 1.35 | 0.76,2.43 | 0.316 |
| | Q4 (NMLR≥ 3.21) | 1.85 | 1.09,3.21 | 0.026* | 1.74 | 0.99,3.12 | 0.056 | 1.46 | 0.81,2.65 | 0.210 |
| | p for trend | | | 0.005* | | | 0.012* | | | 0.066 |
| SIRI | SIRI continuous | 1.32 | 1.14,1.51 | <0.001* | 1.29 | 1.11,1.49 | <0.001* | 1.25 | 1.06,1.46 | 0.006* |
| | SIRI quantile | | | | | | | | | |
| | Q1 (SIRI< 0.8) | Ref | Ref | | Ref | Ref | | Ref | Ref | |
| | Q2 (0.8≤SIRI< 1.17) | 0.79 | 0.42,1.45 | 0.447 | 0.74 | 0.39,1.37 | 0.341 | 0.66 | 0.35,1.24 | 0.195 |
| | Q3 (1.17≤SIRI< 1.69) | 1.36 | 0.79,2.37 | 0.266 | 1.27 | 0.72,2.24 | 0.412 | 1.12 | 0.63,2.00 | 0.703 |
| | Q4 (SIRI≥ 1.69) | 1.64 | 0.97,2.81 | 0.067 | 1.52 | 0.87,2.69 | 0.147 | 1.23 | 0.69,2.23 | 0.479 |
| | p for trend | | | 0.018* | | | 0.044* | | | 0.188 |

Abbreviations: OR, odds ratio; CI, confidence intervals

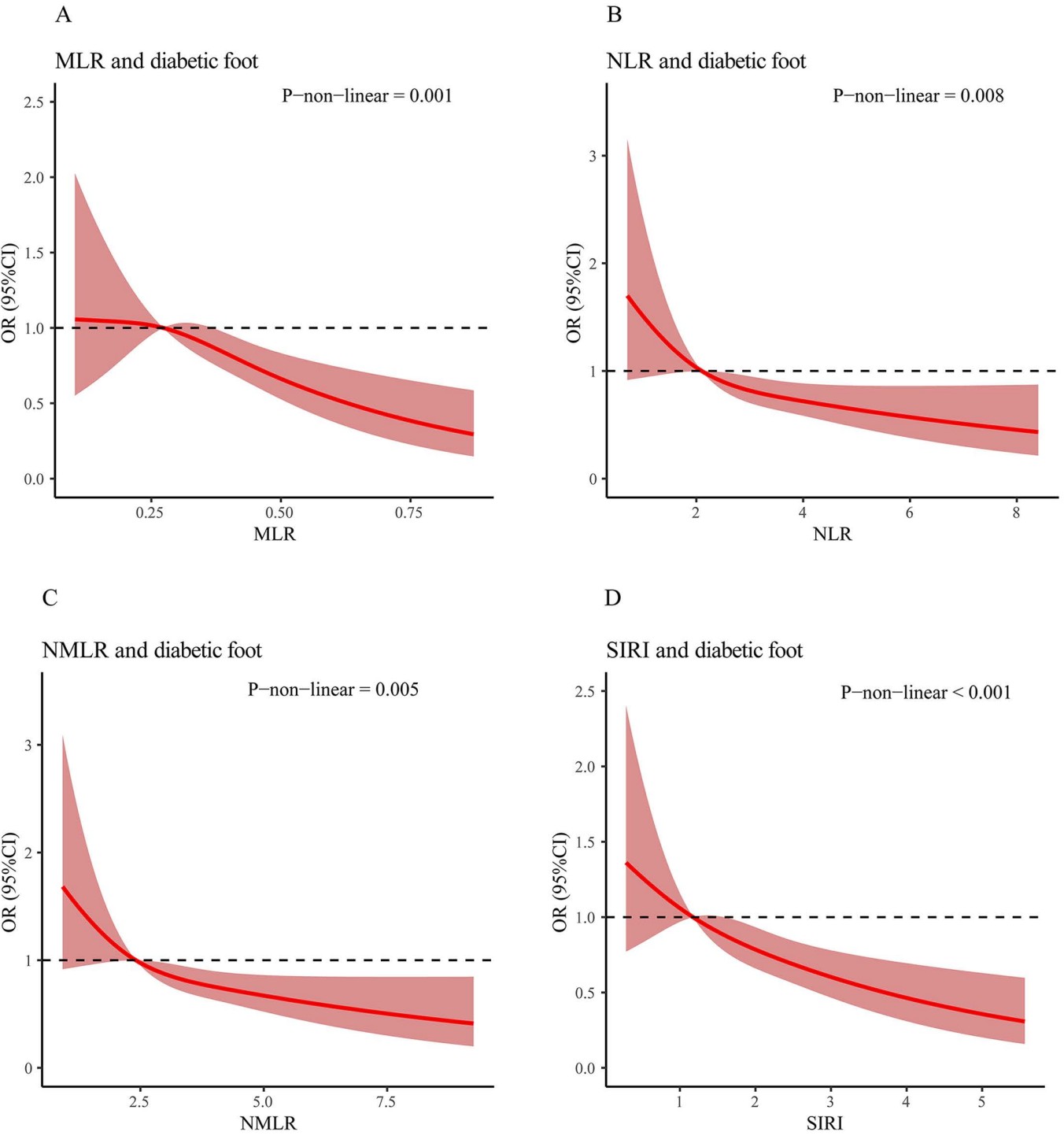

**Fig 2. Relationship between CBC-derived inflammatory markers and diabetic foot RCS curve.**

## Associations of metabolic-related indicators with CBC-derived inflammatory markers and diabetic foot

Table 3 presented the association between CBC-derived inflammatory markers and metabolic-related indicators after multivariable logistic regression. After adjusting for all confounding factors, MLR, NLR, NMLR, and SIRI were positively correlated with RDW and serum creatinine, and negatively correlated with albumin. Specific correlation data are provided below.

(1) CBC-derived inflammatory markers were positively correlated with RDW: MLR (β = 0.26, 95% CI = 0.17, 0.36, P < 0.001), NLR (β = 0.22, 95% CI = 0.13, 0.31, P < 0.001), NMLR (β = 0.23, 95% CI = 0.14, 0.32, P < 0.001), SIRI (β = 0.22, 95% CI = 0.13, 0.31, P < 0.001).

(2) CBC-derived inflammatory markers were negatively correlated with albumin: MLR (β = −0.64, 95% CI = −0.93, −0.34, P < 0.001), NLR (β = −0.56, 95% CI = −0.86, −0.26, P < 0.001), NMLR (β = −0.58, 95% CI = −0.88, −0.29, P < 0.001), SIRI (β = −0.39, 95% CI = −0.68, −0.10, P = 0.009).

(3) CBC-derived inflammatory markers were positively correlated with serum creatinine: MLR (β = 0.29, 95% CI = 0.15, 0.44, P < 0.001), NLR (β = 0.14, 95% CI = 0.02, 0.27, P = 0.028), NMLR (β = 0.15, 95% CI = 0.03, 0.28, P = 0.011), SIRI (β = 0.15, 95% CI = 0.03, 0.28, P = 0.015).

## The mediating effect of metabolic-related indicators

The study found that serum creatinine mediated 12.46% of the association between CBC-derived inflammatory markers and diabetic foot (Fig 3A). In the analysis of albumin, the mediating proportion was 11.33% (Fig 3B). Additionally, we also assessed the mediating effect of other metabolic indicators such as RDW (S1 Table).

## Discussion

Diabetic foot is a serious complication affecting over 1.3 billion people projected to have diabetes by 2050 [10]. Its pathogenesis involves intricate interactions between chronic hyperglycemia, microvascular dysfunction, immune dysregulation,

**Table 3. The associations between CBC-Derived inflammatory markers and metabolic-related indicators in diabetic foot.**

|  | MLR | | | NLR | | | NMLR | | | SIRI | | |
|---|---|---|---|---|---|---|---|---|---|---|---|---|
|  | β | 95% CI | P value | β | 95% CI | P value | β | 95% CI | P value | β | 95% CI | P value |
| RDW |  |  |  |  |  |  |  |  |  |  |  |  |
| Model1 | 0.24 | 0.15,0.32 | < 0.001* | 0.17 | 0.09,0.25 | < 0.001* | 0.18 | 0.10,0.26 | < 0.001* | 0.19 | 0.11,0.27 | < 0.001* |
| Model2 | 0.28 | 0.19,0.38 | < 0.001* | 0.25 | 0.16,0.33 | < 0.001* | 0.25 | 0.17,0.34 | < 0.001* | 0.26 | 0.17,0.35 | < 0.001* |
| Model3 | 0.26 | 0.17,0.36 | < 0.001* | 0.22 | 0.13,0.31 | < 0.001* | 0.23 | 0.14,0.32 | < 0.001* | 0.22 | 0.13,0.31 | < 0.001* |
| ALB |  |  |  |  |  |  |  |  |  |  |  |  |
| Model1 | −0.32 | −0.60,-0.04 | 0.023* | −0.28 | −0.56,-0.00 | 0.048* | −0.3 | −0.58,-0.03 | 0.032* | −0.18 | −0.45,0.09 | 0.196 |
| Model2 | −0.69 | −0.98,-0.39 | < 0.001* | −0.63 | −0.93,-0.34 | < 0.001* | −0.66 | −0.96,-0.37 | < 0.001* | −0.46 | −0.76,-0.18 | 0.002* |
| Model3 | −0.64 | −0.93,-0.34 | < 0.001* | −0.56 | −0.86,-0.26 | < 0.001* | −0.58 | −0.88,-0.29 | < 0.001* | −0.39 | −0.68,-0.10 | 0.009* |
| Cr |  |  |  |  |  |  |  |  |  |  |  |  |
| Model1 | 0.49 | 0.32,0.67 | < 0.001* | 0.22 | 0.10,0.36 | < 0.001* | 0.24 | 0.12,0.38 | < 0.001* | 0.23 | 0.11,0.36 | < 0.001* |
| Model2 | 0.31 | 0.18,0.47 | < 0.001* | 0.17 | 0.05,0.30 | 0.005* | 0.19 | 0.07,0.32 | 0.002* | 0.2 | 0.08,0.33 | 0.001* |
| Model3 | 0.29 | 0.15,0.44 | < 0.001* | 0.14 | 0.02,0.27 | 0.028* | 0.15 | 0.03,0.28 | 0.011* | 0.15 | 0.03,0.28 | 0.015* |

Abbreviations: RDW Red blood cell distribution width, ALB albumin, Cr serum creatinine

Note: Model 1 was the crude model; Model 2 was adjusted by Gender, Age, Race, Edu, PIR, Marry; Model 3 was adjusted by Gender, Age, Race, Edu, PIR, Marry, Drink, Hypertension, Heart failure, CHD

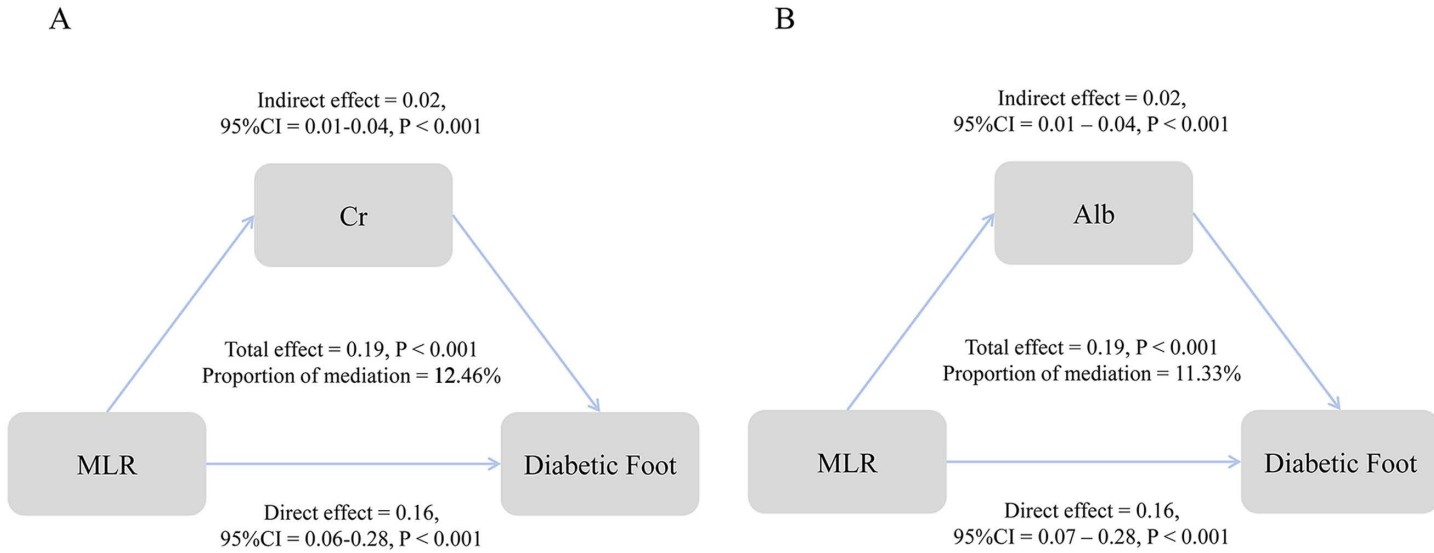

**Fig 3. Mediating effect of creatinine and albumin on the association of CBC-derived inflammatory markers.**

and metabolic disturbances, creating a vicious cycle of impaired wound healing and progressive tissue damage [11]. The complexity of these pathways underscores the urgent need for biomarkers capable of risk stratification and mechanistic insight.

Our research identified CBC-derived inflammatory markers—MLR, NLR, NMLR, and SIRI—as strong predictors of diabetic foot risk, with the risk increasing alongside the quartiles of these markers.In patients with diabetic foot, the inflammatory cell infiltration and mediator release lead to local tissue damage, affecting wound healing [12]. In addition, the inflammatory response may also exacerbate pathological changes and delay wound healing by affecting the nervous and vascular systems [13,14]. Our findings are also supported by previous studies. The increase in MLR and NLR is highly correlated with gestational hyperglycemia, diabetic foot, and non-healing ulcers of the lower limbs [15–17]. And NLR is significantly associated with higher amputation rates and recurrence rates in patients with diabetic foot, with longer duration of antibiotic treatment, longer hospital stays, and longer ulcer healing times [18–20]. Moreover, NLR, MLR, and SIRI were significantly associated with all-cause mortality and cardiovascular disease mortality in American adults with diabetes [21]. Related studies have also confirmed a strong correlation between SIRI and peripheral artery disease in patients with type 2 diabetes [22]. Although there is currently a lack of research on the relationship between NMLR and diabetic foot, it has been confirmed that NMLR is associated with acute myocardial infarction. [23,24]. Statistics show that the probability of acute myocardial infarction in diabetic patients is 11% [25], while other related data can reach 24.1% [26]. Interestingly, our subgroup analysis found that the positive correlation between MLR and diabetic foot was stronger in participants with coronary heart disease.This reminds us to pay attention to diabetic patients with coronary heart disease. We chose the restricted cubic spline (RCS) method, revealing a nonlinear relationship between CBC-derived inflammatory markers and diabetic foot. Due to RCS's unique ability to identify inflection points without overfitting [27], it can model and demonstrate the inflammation of diabetic foot as a pathological nonlinear process transitioning from compensatory, which is not achievable with linear models. This nonlinearity may be attributed to the complex interactions between different components of the immune system, as well as the different stages of inflammation and healing in DFUs [27,28]. During the wound healing process, dysregulation of the immune system can lead to persistent inflammation and delayed healing, ultimately resulting in chronic wounds [29].

Metabolic-related indicators, particularly serum creatinine (12.46% mediating effect) and albumin (11.33% mediating effect), further reveal the comprehensive impact mechanism of the inflammation-metabolic interaction on diabetic foot. First, the elevated serum creatinine reflects declining renal filtration capacity, leading to systemic accumulation of uremic toxins linked to diabetic nephropathy-foot ulcer comorbidity [30,31], the impaired toxin clearance amplifies TLR4/NF-κB signaling in dermal fibroblasts, increasing MMP-9 production and collagen degradation [32]. Then, the uremic metabolites (e.g., indoxyl sulfate) directly inhibit keratinocyte migration and angiogenesis via aryl hydrocarbon receptor activation [33]. In addition, inflammation inhibits the liver's synthesis, leading to hypoalbuminemia [34], which further reduces colloid osmotic pressure, impairs microcirculation flow, and exacerbates tissue hypoxia [35]. The reduction in antioxidant capacity and amplification of oxidative stress—both of which delay epithelial regeneration—ultimately result in poor healing [36], and even a higher risk of amputation [37].

A major advantage of our study is that it is a prospective cohort study that utilizes a large and representative NHANES database, which enhances the generalizability of our findings. Our findings position CBC-derived inflammatory markers as cost-effective tools for diabetic foot risk prediction while elucidating their biological underpinnings in metabolic-immune integration. However, several limitations warrant consideration. While we adjusted for major confounders, residual bias from unmeasured factors like subclinical infections may persist. Self-reported comorbidities, though supplemented by laboratory data, remain vulnerable to recall bias.Future studies should integrate multi-omics profiling to dissect causal pathways and evaluate whether correcting these markers modifies hard endpoints like amputation-free survival. By bridging epidemiological observations with mechanistic insights, this work advances a precision medicine framework for diabetic foot management.

## Conclusions

In summary, our study identifies CBC-derived inflammatory markers as easily accessible predictive indicators of diabetic foot risk, exhibiting a nonlinear pattern that allows for early intervention in high-risk patients. The partial mediating role of metabolic indicators highlights the dual inflammatory-metabolic pathways in the pathogenesis of diabetic foot. Routine CBC-derived inflammatory markers monitoring could enable early risk stratification, while targeting metabolic abnormalities may amplify preventive strategies.

## Supporting information

**S1 Fig. The relationship between baseline characteristics of each subgroup of CBC-derived inflammatory markers and diabetic foot.**
(PDF)

**S1 Table. Analysis of the mediation by Metabolic related indicators of the associations of CBC-derived inflammatory markers.**
(DOCX)

## Acknowledgments

All authors are very grateful for the support of the Foresea Life Insurance Guangzhou General Hospital.

## Author contributions

**Conceptualization:** Yang Zhang, Shumin Zhou.

**Data curation:** Yang Zhang, Shumin Zhou, Xianbin Wang.

**Formal analysis:** Yang Zhang, Shumin Zhou, Xianbin Wang.

**Funding acquisition:** Yang Zhang.

**Investigation:** Yang Zhang, Shumin Zhou, Xianbin Wang.

**Methodology:** Yang Zhang, Shumin Zhou, Xianbin Wang.

**Project administration:** Yang Zhang, Shumin Zhou.

**Resources:** Yang Zhang, Shumin Zhou.

**Software:** Yang Zhang, Shumin Zhou, Haiyan Zhou.

**Supervision:** Yang Zhang, Shumin Zhou, Haiyan Zhou.

**Validation:** Yang Zhang, Shumin Zhou, Xianbin Wang.

**Visualization:** Yang Zhang, Shumin Zhou, Xianbin Wang.

**Writing – original draft:** Yang Zhang, Shumin Zhou.

**Writing – review & editing:** Yang Zhang, Shumin Zhou, Xianbin Wang, Haiyan Zhou.

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
