## [Decision Letter · Decision Letter 0]

PONE-D-25-04532Are metabolic abnormalities the missing link between complete blood count-derived inflammatory markers and diabetic foot?  Evidence from a large population study.PLOS ONE?

Dear Dr. Zhang,

Thank you for submitting your manuscript to PLOS ONE. After careful consideration, we feel that it has merit but does not fully meet PLOS ONE’s publication criteria as it currently stands. Therefore, we invite you to submit a revised version of the manuscript that addresses the points raised during the review process.

**ACADEMIC EDITOR: Minor revision**

We look forward to receiving your revised manuscript.

Kind regards,

Raffaele Vitiello

Academic Editor

PLOS ONE

Journal Requirements:

2. Thank you for stating the following financial disclosure: [This work was supported by Health Science and Technology Project of Guangzhou City, 20241A010119.]

Reviewers' comments:

Reviewer's Responses to Questions

**Comments to the Author**

1. Is the manuscript technically sound, and do the data support the conclusions?

Reviewer #1: Partly

Reviewer #2: Yes

2. Has the statistical analysis been performed appropriately and rigorously?

Reviewer #1: Yes

Reviewer #2: Yes

3. Have the authors made all data underlying the findings in their manuscript fully available?

Reviewer #1: Yes

Reviewer #2: Yes

4. Is the manuscript presented in an intelligible fashion and written in standard English?

Reviewer #1: Yes

Reviewer #2: Yes

Reviewer #1: Your text is scientifically solid, well-structured, and methodologically sound. However, improving the conceptual flow by enhancing transitions, clarifying methodological choices, and adding brief theoretical interpretations would strengthen the overall impact of your analysis.

Reviewer #2: In general, the article seems to be a useful, great value input. I understand that the objective of this work is to make a comprehensive article with all of the available information regarding CBC-derived inflammatory markers and diabetic foot.

The Conclusion section is really simplistic. Authors should rewrite this section with more scientific conclusions related to the discussion section.

Maybe it is possible to emphasize the differences, and the statistical significance of the results from the 3 Models ( Model 1-2-3) you have used in the article.

**Do you want your identity to be public for this peer review?** For information about this choice, including consent withdrawal, please see our Privacy Policy

Reviewer #1: No

Reviewer #2: No

---

## [Author Response · Author response to Decision Letter 1]

13 Apr 2025

Dear Raffaele Vitiello and reviewers:

Thank you very much for your review of my article and for the valuable feedback from the reviewers. The suggestions provided are very helpful for improving our paper. We take every comment from the reviewers seriously and have promptly engaged in discussions within our team to revise the manuscript. I hope the reviewers can feel our sincerity and effort, as we are eager to present a valuable article that gains recognition from the reviewers and fellow scholars.

We have separately uploaded a document titled "Response to Reviewers," in which we have highlighted the reviewers' comments in "bold" for convenience and marked our responses with underlines (please download the file to view).

Below are our responses to each comment.

Abstract:

o While generally well-structured, it should be more concise. The description of statistical methods could be streamlined to avoid overwhelming the reader.

Response:

The statistical descriptions in the original manuscript were indeed overly detailed. We have streamlined this section, focusing on the following aspects:

1.Removed redundant statistical terminology (e.g., "covariates," "potential mechanism").

2.Consolidated analytical approaches into active voice (e.g., "assessed," "supplemented").

3.Simplified the mediation analysis description.

After revisions, the abstract word count was reduced by 28% (original: 267 words → revised: 191 words).

(Page2, lines 23-43)

o Conclusion should reinforce the clinical significance of the findings rather than simply restating them.

Response:

We have revised this section and enhanced clinical conclusions, focusing on the following aspects:

1.Added practical implications: "accessible predictors," "early risk stratification," "amplify preventive strategies".

2.Emphasized translational value: "dual pathways," "routine monitoring".

3.Transitioned from restating results ("nonlinear relationship") to clinical actionability.

(Page2, lines 39-43)

Introduction

o The background is informative but could be more engaging. Consider starting with a strong statement on the burden of diabetic foot rather than statistics.

Response:

Per the reviewers' suggestions, we have implemented the following revisions:

1.Added statements emphasizing disease burden in the introductory section prior to statistical data presentation.

2.Reframed amputation/mortality rate data as consequences of clinical challenges.

3.Introduced the "early risk stratification tool" terminology to foreground the research objectives.

(Page3, lines 49-58)

o The rationale for the study could be framed more clearly. While the study highlights gaps in previous research, a more explicit statement on why this study is novel would improve the introduction.

Response:

Per the reviewers' suggestions, we have implemented the following revisions:

1.Explicitly named limitations of prior studies ("isolated markers," "no population-level research").

2.Highlighted novelty through "three key innovations" section.

3.Connected metabolic mediation hypothesis to biological plausibility ("hypoalbuminemia and elevated creatinine").

(Page4, lines 76-86)

Materials & Methods

o The study design is well-detailed but somewhat dense, sections could be made more concise to enhance readability (e.g., reducing redundancy in technical terms).

Response:

We have rigorously implemented the reviewers' suggestions by removing redundant content and streamlining structural elements, with principal modifications as follows:

1.Consolidated NHANES methodology.

2.Simplified covariate descriptions (e.g., merged demographic/laboratory data sources).

3.Removed explanatory paragraphs about RCS/mediation theory (retained only analytical application).

(Page4-7, lines 88-147)

Discussion

o The discussion could benefit from a brief interpretation of why these metabolic markers mediate the relationship between CBC-derived inflammatory markers and diabetic foot. Do these biomarkers reflect a specific pathophysiological mechanism? A short statement on this would improve the theoretical depth of the analysis.

Response:

Through the reviewers' suggestions, we have realized that the Discussion section needs to strengthen the depth and connotation. Therefore, we have decided to consolidate the original six paragraphs into four, conducting a more in-depth discussion layer by layer.

We have summarized the four paragraphs to ensure that the reviewers can clearly understand our ideas:

1.The first paragraph briefly describes the severity of diabetic foot, emphasizing the complexity of its pathological mechanisms.

2.The second paragraph mainly discusses the relationship between inflammation and diabetic foot, along with its pathophysiological mechanisms. It then explains the advantages of choosing restricted cubic splines (RCS).

3.The third paragraph emphasizes that this study found metabolic abnormalities to be closely related to diabetic foot and continues to analyze the comprehensive impact mechanism of inflammation-metabolism interactions on diabetic foot.

4.The fourth paragraph provides a brief summary, linking the findings of this study to clinical practice, and proposes that CBC-derived inflammatory markers can be positioned as tools for predicting the risk of diabetic foot. It then emphasizes the bias mechanisms and suggests directions for future research.

(Page14-16, lines 228-286)

The above content will serve as the basis for the following three responses.

o Consider adding a brief concluding remark summarizing the implications of the results. This would help tie together the statistical findings with their broader clinical or epidemiological significance.

Response:

We have described the relevant content in the fourth paragraph of the Discussion

The fourth paragraph provides a brief summary, linking the findings of this study to clinical practice, and proposes that CBC-derived inflammatory markers can be positioned as tools for predicting the risk of diabetic foot. It then emphasizes the bias mechanisms and suggests directions for future research.

(Page16, lines 276-286)

o The section on the Restricted Cubic Spline (RCS) method could briefly explain why it was chosen over other non-linear modeling approaches. What specific advantages does it offer in this study?

Response:

We have described the relevant content in the second paragraph of the Discussion

The second paragraph mainly discusses the relationship between inflammation and diabetic foot, along with its pathophysiological mechanisms. It then explains the advantages of choosing restricted cubic splines (RCS).

(Page14-15, lines 235-262)

o Going deeper into the correlation between the most significant analyzed parameters and their relationship with diabetic foot could enrich the discussion, also specifying the mechanisms of possible biases (correlation with other concomitant diseases) to provide a starting point for future similar analyses.

Response:

1.We adopted a layered approach for the discussion, enriching the content. we discussed the impact of inflammation on diabetic foot in the second paragraph, and then in the third paragraph, we delved deeper into the comprehensive mechanisms of the inflammation-metabolism interaction on diabetic foot.

2.The fourth paragraph provides a brief summary, linking the findings of this study to clinical practice, and proposes that CBC-derived inflammatory markers can be positioned as tools for predicting the risk of diabetic foot. It then emphasizes the bias mechanisms and suggests directions for future research.

(Page14-16, lines 235-286)

Conclusion

o Emphasize clinical implications of findings can make a more impactful conclusion.

Response:

We have rewritten it to emphasize the clinical significance of the research findings.

(Page17, lines 288-294)

Additional Supplementary Information on “Journal Requirements”

1. Formatting Requirements

Response:

We have modified and standardized the format of the first-level headings according to the formatting requirements. We have adjusted the "Supporting information" to be placed after the "References" as required.

(Page3-4, lines 66-67) DFUs → Diabetic Foot Ulcers(DFUs)

(Page4, lines 88) Materials and Methods → Materials and methods

(Page5, lines 89) Study Design and Population → Study design and population

(Page6, lines 127) Statistical Analysis → Statistical analysis

(Page22, lines 411-415) We have adjusted the "Supporting information" to be placed after the "References" as required.

2.Role of Funder statement

Response:

We have added the role of funder statement in the original Cover letter and resubmitted it in the submission system. The added content is highlighted in bold in the cover letter, as follows:

“It is important to emphasize that this work was supported by the Health Science and Technology Project of Guangzhou City, 20241A010119. The funders had no role in study design, data collection and analysis, decision to publish, or preparation of the manuscript.”

3. References

Response:

We have updated the references according to the revised content of the article. We realized that the initial submitted manuscript did not include DOIs for the references, and we have added the DOIs as required, while also specifically verifying that none of the cited references have been retracted.

Special note: Reference number 26 is “Alwakeel JS, Al-Suwaida A, Isnani AC, Al-Harbi A, Alam A. Concomitant macro and microvascular complications in diabetic nephropathy. Saudi J Kidney Dis Transpl. 2009 May;20(3):402-09. PMID: 19414942.” We have verified that this paper does not have a DOI, which may be related to the fact that the journal did not adopt DOIs in 2009. We have used the PMID of the paper as a substitute. Furthermore, the journal is a legitimate academic journal with an ISSN number (1319-2442), and the paper has certain academic citation value.

(Page17-22, lines 301-409)

Thank you again for your valuable suggestions on our manuscript.

Sincerely,

Yang Zhang

---

## [Decision Letter · Decision Letter 1]

Are metabolic abnormalities the missing link between complete blood count-derived inflammatory markers and diabetic foot?  Evidence from a large population study.

PONE-D-25-04532R1

Dear Dr. Zhang,

We’re pleased to inform you that your manuscript has been judged scientifically suitable for publication and will be formally accepted for publication once it meets all outstanding technical requirements.

Kind regards,

Raffaele Vitiello

Academic Editor

PLOS ONE

Additional Editor Comments (optional):

Reviewers' comments:

Reviewer's Responses to Questions

**Comments to the Author**

Reviewer #1: All comments have been addressed

Reviewer #2: All comments have been addressed

2. Is the manuscript technically sound, and do the data support the conclusions?

Reviewer #1: Yes

Reviewer #2: Yes

3. Has the statistical analysis been performed appropriately and rigorously?

Reviewer #1: Yes

Reviewer #2: Yes

4. Have the authors made all data underlying the findings in their manuscript fully available?

Reviewer #1: Yes

Reviewer #2: Yes

5. Is the manuscript presented in an intelligible fashion and written in standard English?

Reviewer #1: Yes

Reviewer #2: Yes

Reviewer #1: (No Response)

Reviewer #2: I appreciated the improvements to the article, based on the comments. In my opinion the conclusion could have been expanded even more, but I give my consent to its publication

**Do you want your identity to be public for this peer review?** For information about this choice, including consent withdrawal, please see our Privacy Policy

Reviewer #1: No

Reviewer #2: No

---

## [Editor Report · Acceptance letter]

PONE-D-25-04532R1

PLOS ONE

Dear Dr. Zhang,

I'm pleased to inform you that your manuscript has been deemed suitable for publication in PLOS ONE. Congratulations! Your manuscript is now being handed over to our production team.

Kind regards,

on behalf of

Dr. Raffaele Vitiello

Academic Editor

PLOS ONE